# Radiologic versus Segmentation Measurements to Quantify Wilms Tumor Volume on MRI in Pediatric Patients

**DOI:** 10.3390/cancers15072115

**Published:** 2023-04-01

**Authors:** Myrthe A. D. Buser, Alida F. W. van der Steeg, Marc H. W. A. Wijnen, Matthijs Fitski, Harm van Tinteren, Marry M. van den Heuvel-Eibrink, Annemieke S. Littooij, Bas H. M. van der Velden

**Affiliations:** 1Princess Máxima Center for Pediatric Oncology, 3584 CS Utrecht, The Netherlands; 2Wilhelmina Children’s Hospital, University Medical Center Utrecht, 3584 EA Utrecht, The Netherlands; 3Image Sciences Institute, University Medical Center Utrecht, Utrecht University, 3584 CX Utrecht, The Netherlands

**Keywords:** Wilms tumor, pediatric oncology, volume measurements, MRI, deep learning

## Abstract

**Simple Summary:**

Volume measurements are important in tumor evaluations of children with a Wilms tumor. Current volume measurements might not be accurate. Our study had two aims. Our first aim was to assess whether manual segmentation of MRI can accurately quantify the volume of Wilms tumors. Our second aim was to show if manual segmentation can be automated using deep learning. We compared radiological-based and manual segmentation-based measurements. Next, we developed an automated segmentation method. Radiological measurements underestimate the actual tumor volume by about 10% irrespective of tumor size. Deep learning can potentially be used to replace manual segmentations in volume measurements.

**Abstract:**

Wilms tumor is a common pediatric solid tumor. To evaluate tumor response to chemotherapy and decide whether nephron-sparing surgery is possible, tumor volume measurements based on magnetic resonance imaging (MRI) are important. Currently, radiological volume measurements are based on measuring tumor dimensions in three directions. Manual segmentation-based volume measurements might be more accurate, but this process is time-consuming and user-dependent. The aim of this study was to investigate whether manual segmentation-based volume measurements are more accurate and to explore whether these segmentations can be automated using deep learning. We included the MRI images of 45 Wilms tumor patients (age 0–18 years). First, we compared radiological tumor volumes with manual segmentation-based tumor volume measurements. Next, we created an automated segmentation method by training a nnU-Net in a five-fold cross-validation. Segmentation quality was validated by comparing the automated segmentation with the manually created ground truth segmentations, using Dice scores and the 95th percentile of the Hausdorff distances (HD95). On average, manual tumor segmentations result in larger tumor volumes. For automated segmentation, the median dice was 0.90. The median HD95 was 7.2 mm. We showed that radiological volume measurements underestimated tumor volume by about 10% when compared to manual segmentation-based volume measurements. Deep learning can potentially be used to replace manual segmentation to benefit from accurate volume measurements without time and observer constraints.

## 1. Introduction

Wilms tumor is one of the most prevalent malignant solid tumors in pediatric patients. With a combination of neoadjuvant chemotherapy, surgery, and post-operative chemotherapy survival up to 90% can be achieved [1,2]. Throughout the treatment process, quantifying tumor volume on magnetic resonance imaging (MRI) or computed tomography (CT) is important for several reasons. In the International Society of Paediatric Oncology–Renal Study Tumor Group (SIOP-RTSG) UMBRELLA guidelines MRI is preferred because of the absence of a radiation dose and the excellent soft tissue contrast [3]. First, tumor volume can be used as an indication for therapy response and risk stratification, as a tumor volume of >500 mL after neoadjuvant chemotherapy for specific Wilms tumor types places these patients in a higher risk group (see also Appendix A) [3]. Next, the UMBRELLA guidelines indicate nephron-sparing surgery (NSS) only for tumors <300 mL at diagnosis in patients with a unilateral tumor without a predisposing (tumor) syndrome [4,5]. Last, tumor volume has been suggested to be predictive of patient outcomes [5,6,7]. Hence, tumor volume measurements are important for Wilms tumor management.

In current good clinical practice, review radiologists assess tumor volume on MRI or CT by measuring the largest diameter of the tumor in three directions. This method is based on the assumption that the tumor is an ellipsoid shape, which does not always reflect the true tumor shape [8,9,10]. Volume measurement by manual delineation of the tumor on MRI (i.e., segmentation) does not require this assumption. The extent of error in volume measurements in Wilms tumor introduced by the assumption of ellipsoid tumor shape measurements remains unknown.

Although manual segmentation might be more accurate for volume measurements, the process is time-consuming and intra- and interobserver-dependent [11]. Automated segmentation, e.g., using deep learning, has the potential of utilizing more precise manual segmentation-based volume measurements while limiting the time and observer constraints [12]. The aim of this study was twofold. Our first aim was to assess whether manual segmentation of MRI can accurately quantify the volume of Wilms tumors. Our second aim was to prove if manual segmentation can be automated using deep learning.

## 2. Materials and Methods

### 2.1. Patients

A total of 45 consecutive Wilms tumor patients, 0–18 years of age and treated in the Princess Máxima Center for pediatric oncology during the period of July 2018 to November 2020 were retrospectively included. All patients signed an informed consent form. Patients were excluded when imaging was incomplete (*n* = 4) or when no informed consent for the SIOP-UMBRELLA study was present (*n* = 1).

All but five patients were treated with pre-operative chemotherapy according to the SIOP-RTSG 2001 or SIOP-RTSG 2016-UMBRELLA protocol [3]. Tumor type and stage were assessed on the resection specimen using hematoxylin and eosin (H&E) staining, and additional immunohistochemistry (IHC), according to international SIOP-RTSG standards [13]. All cases underwent rapid pathology review.

### 2.2. Magnetic Resonance Imaging

Standard clinical MRI scans were performed and centrally reviewed at diagnosis and before surgery in accordance with the SIOP-RTSG 2016-UMBRELLA protocol [5]. The imaging was performed on a 1.5T scanner (Ingenia; Philips Medical Systems, Best, The Netherlands), and included a 3D T_2_-weighted scan and a fat-suppressed T_1_-weighted scan with and without intravenous contrast (Gadovist, Bayer Pharma, Berlin, Germany, 0.1 mmol/kg body weight) (Table 1).

### 2.3. Tumor Volume Measurement

We used manual segmentation as reference volume measurements as it is the closest approximation for true tumor volume on MRI. Subsequently, we compared the radiological tumor volume measurement with the manual segmentation-based tumor volume measurement. Next, we compared the deep learning-based segmentations with the manually created ground truth segmentation.

#### 2.3.1. Radiological Tumor Volume Measurements

The largest tumor diameters were measured in three directions by the pediatric radiologist, from which the volume of an ellipsoid was calculated (i.e., the radiological tumor volume).

#### 2.3.2. Manual Segmentation-Based Tumor Volume Measurements

Tumors were manually delineated (i.e., segmented) on the images by three independent observers under supervision of one pediatric radiologist (AL, 13 years of experience with pediatric MRI). Patients were randomly assigned to each observer. A segmentation protocol was developed together with the reviewing radiologist. The manual delineation was performed in 3DSlicer (version 4.11) using a preset protocol [14]:Manual indication of tumor and background on each fourth slice of the postcontrast T_1_-weighted scan, due to the best tumor and kidney contrast in this sequence.Initial tumor segmentation using the 3DSlicer algorithm “grow from seeds”, which is a 3D volume growing algorithm. After this step, each pixel was assigned either the label tumor or background.Because of the difference in in-slice resolution and slice thickness, the segmentation was reformatted from the T_1_-weighted image to the T_2_-weighted image using 3DSlicer’s inbuild function. These labels were extensively checked and manually corrected if needed.

Tumor volume was calculated by multiplying the number of voxels in the tumor segmentation with the voxel size in mL.

#### 2.3.3. Deep Learning-Based Tumor Volume Measurements

Tumor segmentation was automated using the deep learning application nnU-Net, which is a self-adapting framework for U-net-based segmentation [8,15]. Input to nnU-Net were full 3D T_2_-weighted images, output was binary segmentation maps. Patients were randomly split into 5 groups for 5-fold cross-validation [16]. This means that every patient is used as an independent test once and as part of the training data five times. For each fold, a separate nnU-Net training was performed, after which the results of each test group were combined for the final analysis. Differences in baseline characteristics between folds were tested using the chi-squared test to confirm an unbiased split.

A connected component analysis was performed on the output of the nnU-Net segmentation. In unilateral cases, only the biggest connected component was used as tumor segmentation in the next analysis steps. In bilateral cases, the components were visually assessed and manually coupled with the correct tumor label. Figure 1 shows an overview of the deep learning workflow.

### 2.4. Statistical Analysis

To compare the radiological tumor volume measurement with the manual segmentation-based tumor volume measurement, each tumor was analyzed separately. Agreement was assessed using Bland–Altman analyses [17,18]. Tumors were divided into three groups to compare the absolute difference between the groups and the percentage-wise differences.

To assess the performance of the automated deep learning-based tumor volume measurement, automated segmentations were compared to the manual reference segmentations using the Dice similarity coefficient (Dice) and the 95th percentile of the Hausdorff distance (HD95) [19]. The Dice score corresponds with overlay of two segmentation methods. A Dice score of 1.0 is a perfect overlay. The HD95 represents the 95th percentile of the maximum distances between points in border of the two segmentations of the same tumor. A higher HD95 represents a worse comparison. The performance was assessed for all the five independent test sets of the 5-fold cross-validation combined. Differences in segmentation outcomes between different subgroups were tested for statistical significance using Kruskal–Wallis and Mann–Whitney tests. *p*-values < 0.05 were considered statistically significant. Statistical analyses were performed using Python 3.8 and SciPy 1.7.1.

## 3. Results

### 3.1. Patients

Of the 45 patients, 22 were male (49%), and 23 were female (51%). The median age at diagnosis was 39 months (range: 7–109 months) (Table 2). In six patients (13%), bilateral disease was present. One patient (2.2%) had two tumors in one kidney. In total, 52 tumors were separately analyzed (Table 3). The most common histological subtypes consisted of mixed type (*n* = 13) or regressive (*n* = 14) type Wilms tumor. Five tumors, all in bilateral patients, were not surgically resected. Four of them were not resected because the lesion was a radiological suspect for a nephrogenic rest. The fifth lesion was not resected because the patient died before surgery of the second kidney was pursued. For these tumors, histological tumor type could not be determined.

#### 3.1.1. Tumor Volume Measurements

The median tumor volume for manual segmentation-based measurements was 200 mL (range: 0.78–1757 mL (Figure 2). The median tumor volume in patients with bilateral disease was 8 mL (range: 0.78–1105 mL), in contrast with a median tumor volume of 307 mL (range: 3–1757 mL) for unilateral tumors.

#### 3.1.2. Radiological versus Manual Segmentation-Based Tumor Volume Measurements

The median difference between radiological tumor volume measurements and segmentation-based tumor volume measurements was 14 mL (range: −81 mL–+175 mL) (Figure 3). This corresponds to a median underestimation of 10% (range: −28.6%–+52.6%). In 38 of 52 tumors (73%), the clinically used radiological tumor volume measurements had a smaller volume compared to the manual segmentation-based volume measurements. For two tumors (4%), the volume measurements yielded, the same result when rounded to the nearest mL. In nineteen tumors (23%), radiological tumor volume measurements had a larger tumor volume.

When the tumors were divided into three groups based on their manual volume (0–300, 300–500 mL, and >500 mL) there was a significant increase in the absolute difference between the radiological and manual segmentation volume. However, the difference in the percentage of the volume was the same for the three groups (please refer to Table 4).

#### 3.1.3. Deep Learning-Based Segmentation

When comparing the manual and automated segmentation for each segmented tumor (*n* = 52), the median non-zero Dice was 0.90 (range: 0.03–0.97) (0.89 for fold 1, 0.9 for fold 2, 0.90 for fold 3, 0.95 for fold 4, 0.89 for fold 5, folds did not differ significantly). The median Hausdorff distance was 7.2 mm (5.5 for fold 1, 7.2 for fold 2, 11.3 for fold 3, 7.0 for fold 4, and 8.0 for fold 5. There was no significant difference between the different folds). Segmentations for the patient with both the highest and lowest Dice are depicted in Figure 4. The median HD95 was 7.2 mm (range: 1.7–27 mm).

Deep learning-based segmentation failed to segment three tumors (6%). Each of the missed tumors belonged to the subset of patients with bilateral disease. The missed tumors were relatively small: the manual segmentation-based volumes of the missed tumors were 1.2, 1.9, and 5.7 mL.

Dice scores were significantly lower in patients with bilateral disease. In these patients, the median Dice was 0.79, whereas this was 0.91 for patients with unilateral disease. The median HD95 was 9.3 mm compared with 7.0 mm, which was not significant (*p* = 0.43).

Tumors smaller than 300 mL had a significantly lower median Dice of 0.75 compared with tumors between 300 and 500 mL with a median Dice of 0.90 and above 500 mL with a median Dice of 0.92. The Dice did not significantly differ between tumors of 300 to 500 mL and those above 500 mL (*p* = 0.19). HD95 did not depend on tumor volume (*p* = 0.99).

No significant difference was observed among the various histological subtypes of Wilms tumors in Dice scores (*p* = 0.11) and HD95 (*p* = 0.85). In addition, no difference was found in the Dice score (*p* = 0.49) and HD95 (*p* = 0.55) between patients who received neoadjuvant chemotherapy and those who did not, nor between patients with an age higher or lower than the median age (Dice *p* = 0.057, HD95 *p* = 0.27).

## 4. Discussion

Our results indicate that conventional radiological tumor volume assessment underestimates tumor volume in the majority of cases when compared to manual segmentation. We further showed that deep learning can be used to replace time-consuming manual segmentation.

Radiological tumor volume measurements assume that the shape of the tumor is an ellipsoid, whereas manual segmentation-based volume measurements are based on the number of tumor voxels. This is a direct method of measuring without assumptions concerning the shape of the tumor. We indeed found an average underestimation of 10% for radiological tumor volume measurements when compared to manual segmentation-based volume measurements.

The literature in adult oncology supports our findings that geometry-based measurements are less accurate. For example, Colvin et al. showed that ellipsoid volume measurements overestimate total prostate volume [10]. Furthermore, Tiruman et al. showed that ellipsoid-based volume measurements were incorrect when the shape of a tumor was irregular [8]. Müller et al. found an average underestimation of 22% in Wilms tumor volumes when measured as an ellipsoid compared to human expert annotations (*n* = 17 patients) [11]. Our paper, which is the largest to date regarding the number of Wilms tumor patients, also shows an underestimation and provides a deep learning framework to accurately quantify Wilms tumor volume. Together, this inaccuracy of geometry-based tumor volume measurements is consistent with our findings.

Clinically, two volume thresholds are of importance. First, the Society of Paediatric Oncology–Renal Study Tumor Group (SIOP-RTSG) UMBRELLA guidelines consider localized intermediate risk (non-stromal and non-epithelial) stage II and III tumor types high risk when the measured pre-operative volume after chemotherapy is >500 mL [5]. In our cohort, two patients (4.5%) had radiological volume measurements of <500 mL, but >500 mL when using manual segmentation-based volume measurements, which would place these patients in a higher risk group. Our research suggests that if segmentation-based volume measurements become clinically implemented, children can be treated more accurately. Obviously, the prognostic value of volume-based assessment by manual or automated assessment needs to be confirmed based on post hoc analyses before implementation in clinical practice.

The other category of interest is the children eligible for nephron-sparing surgery (NSS), which is currently only considered to be safe for patients with a unilateral tumor with a volume of <300 mL on the diagnostic MRI scan. Two patients (4.5%) were eligible for NSS when measured by the radiologist but crossed the threshold of 300 mL tumor volume when measured manually segmentation-based. Although these results were based on post-chemo MRI scans, we expect that the same applies to diagnostic MRI scans. Together, these results show that the method of tumor volume measurement has potential clinical consequences in clinical decision-making.

In addition to volume measurements, other factors are of importance in staging and risk stratification of Wilms tumor. Ma et al. showed that they could accurately predict WT stage 1 on CT research on the automatic staging of WT on MRI is needed [20].

Although manual segmentation-based volume measurements seem to be more accurate, this process is time-consuming, ranging from one to multiple hours, and is observer dependent [11,21,22]. Automating segmentation using deep learning would limit these constraints. Deep learning-based segmentation was accurate in most of the cases, with a median Dice of 0.90. The highest Dice was 0.97, an almost perfect overlap between the manual and automated segmentation. In contrast, the lowest overall Dice score was 0.26 for one side and 0 for the other side, which is insufficient for future clinical applications. Figure 4 shows both of these patients together. The first thing that stands out is the big difference in tumor volume between these patients. Our analysis indeed showed a significantly lower Dice score for tumors below <300 mL. This can partially be explained by the Dice being a volume-dependent measurement [23], but the lower Dice scores were not merely an artifact of the evaluation metric. Several explanations may exist: first, bilateral tumors were typically smaller (median 8 mL) than unilateral tumors (median 307 mL) and the number of bilateral tumors was smaller in our training set (*n* = 6). Therefore, our segmentation method might be biased toward large unilateral tumors. We will investigate this in future studies in a larger dataset with an increased number of small unilateral and bilateral tumors separately. Second, the appearance of the small tumors can be visually different, which is exemplified in Figure 4. Both patients in Figure 4 had a tumor of mixed type, but visually these tumors are not the same. The unilateral tumor with the high Dice score is big, hyperintense, and clearly separable from the surrounding kidney tissue. In contrast, the two tumors in the bilateral case are small and of roughly the same intensity as the surrounding kidney tissue. For all other clinical characteristics, we did not find a significant difference in the Dice score or the 95th percentile of the Hausdorff distances.

The differences between manual segmentation and radiological volume measurements may apply to many different types of tumors. We decided to conduct this research on WT patients first because of the importance of volumetric assessment during treatment. Furthermore, in our hospital the solid tumor diagnosed the most is WT. So, this new method of volume measurements will have clinical implications rather quickly. Although the automated segmentation method was developed specifically for Wilms tumor patients, the general methodology can easily be extrapolated to other tumor types. We hypothesize that our conclusion about the radiological volume measurements holds across tumor types. The developed automated segmentation method can easily be retrained for other tumor types, both in the pediatric and adult populations. In future research, we will extend our framework to other pediatric tumor types such as neuroblastoma. This study had some limitations. First, we had a limited sample size. This is, however, the largest unselected national MRI cohort to date including all pediatric Wilms tumor patients in the Netherlands in the period of 2018–2020 [2,24]. Second, although we divided the tumors over three observers for segmentation, the population of tumors and the techniques are derived from a single-institution study. We plan to validate our findings in a large prospective multicenter cohort study.

We showed that clinically used volume measurements in Wilms tumor are not accurate. Changing the method of volume measurements might impact clinical decision-making. Further research has to be conducted to investigate the clinical impact of our findings.

## 5. Conclusions

Our results show that currently used geometry-based radiological tumor volume measurements by review radiologists underestimate tumor volume in Wilms tumors compared to manual segmentation. This underestimation was about 10%, independent of tumor size. A change in volume measurement technique may have consequences for the stratification of treatment in individual children with a Wilms tumor and thus have clinical implications.

Deep learning shows the potential to replace manual segmentation in this process to utilize the benefits of segmentation-based volume measurements while limiting time and observer constraints. This method of volume measurements using manual or semi-automated segmentation may be extrapolated to other (pediatric) tumor types.

## Figures and Tables

**Figure 1 cancers-15-02115-f001:**
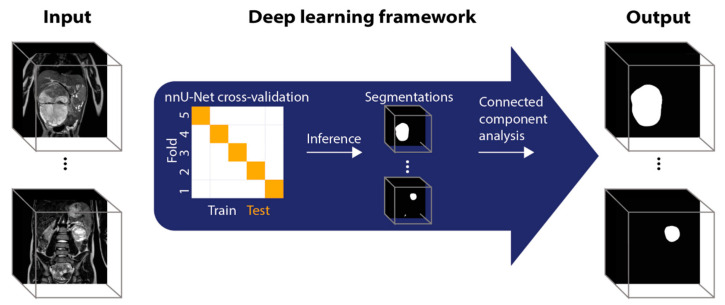
Schematic overview of the deep learning image analysis workflow. All analyses are performed in 3D. Images in the figure are 2D schematic representations of 3D volumes.

**Figure 2 cancers-15-02115-f002:**
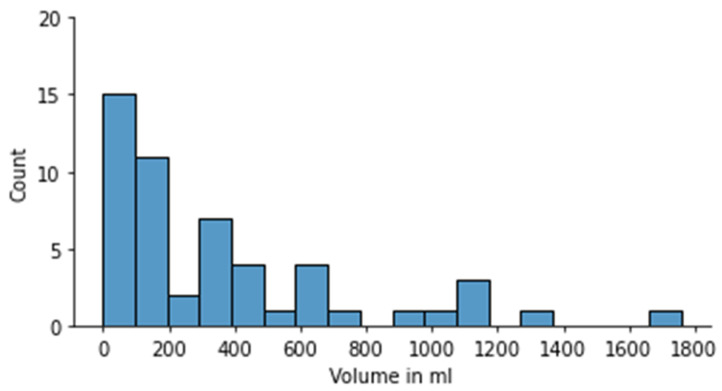
Volumes as assessed by review radiologist.

**Figure 3 cancers-15-02115-f003:**
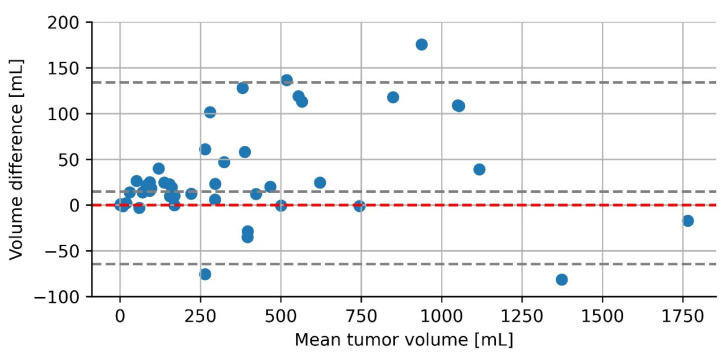
Radiological vs. manual-based tumor volumes. The volume difference plotted on the y-axis was calculated by extracting the radiological tumor volumes from the manually measured reference volume. The red line corresponds to no difference with the manually determined reference volume. Because radiological tumor volume was subtracted from the reference volume, a point above the red line presents an underestimation of tumor volume by the radiologist while points below the line represent an overestimation of tumor volume. In grey, the median and 2.5–97.5 percentile are plotted.

**Figure 4 cancers-15-02115-f004:**
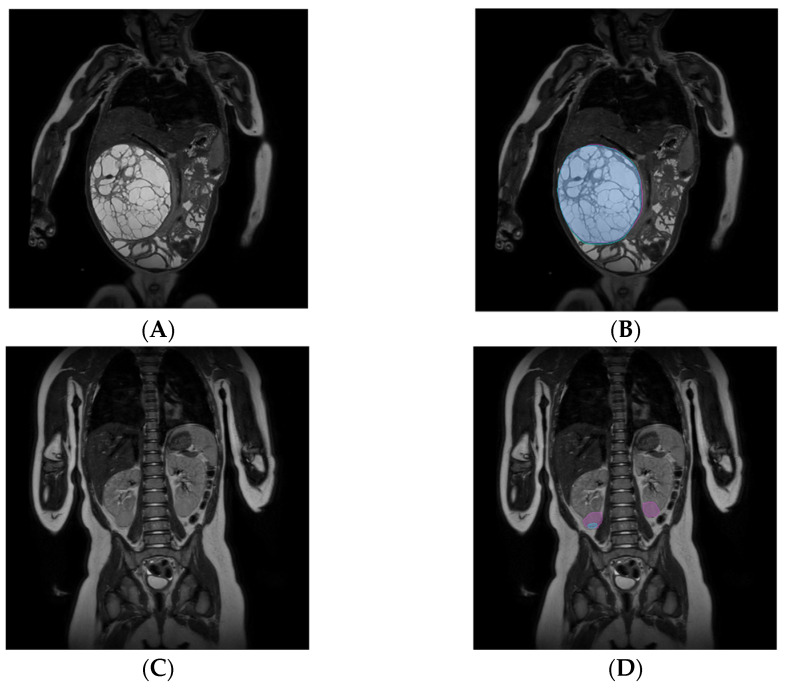
Examples of automated Wilms tumor segmentation. Patients with the highest Dice score (top row (**A**): MRI, (**B**): MRI with measurements; Dice score = 0.97) and the lowest overall Dice score (bottom row (**C**): MRI, (**D**) MRI with measurements; Dice score = 0.26 for the right kidney tumor, Dice score = 0 for the left kidney tumor as it was missed). The reference manual segmentation is shown in purple, and the deep learning segmentation is projected on the manual segmentation in blue. Note that in (**B**) these regions have an almost complete overlap.

**Table 1 cancers-15-02115-t001:** MRI imaging parameters.

Parameters	T_1_-Weighted with Fat Suppression	3D T_2_-Weighted
Sequence type	Gradient Echo	Turbo Spin Echo
Repetition time (ms)	5.5	459
Echo time (ms)	2.7	90
Flip angle	10°	90°
Slice thickness (mm)	3.0	1.15
Voxel spacing (mm)	0.74 × 0.74 mm^2^	0.83 × 0.83 mm^2^

**Table 2 cancers-15-02115-t002:** Baseline characteristics (*n* = 45). Numbers are count (percentage) unless otherwise specified.

Characteristic		Value
Median age at diagnosis in months (min–max)		39 (7–109)
Gender	Male Female	22 (49%) 23 (51%)
Tumor localization	Bilateral Left Right	6 (13%) 16 (36%) 23 (51%)

**Table 3 cancers-15-02115-t003:** Overview of analyzed tumors (*n* = 52). Numbers are count (percentage) unless otherwise specified. For 5 tumors, histological type was unknown because these tumors were not resected.

Characteristics		Value
Histological tumor type	Regressive Non-regressive - Mixed - Stromal - Epithelial - Blastemal Diffuse anaplasia Completely necrotic Nephrogenic rest Unknown	14 (27%) 13 (25%) 9 (17%) 2 (4%) 1 (2%) 5 (10%) 2 (4%) 1 (2%) 5 (10%)
Median radiological volume [mL] (range)		215 (0.68–1774)

**Table 4 cancers-15-02115-t004:** Differences between radiological and manual segmentation volume in absolute numbers and percentages.

Volume Tumor	Absolute Difference (Mean)	*p*-Value	Percentage Difference (Mean)	*p*-Value
0–300 mL	5.6	0.01	11.9	0.95
300–500 mL	21.5		9.1	
>500 mL	70.2		9.2	

## Data Availability

The datasets analysed during the current study are not publicly available due to patient privacy. Upon reasonable request to the corresponding author the raw data can be made available, however the imaging itself will not be shared.

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
