# Peer review of "Radiologic versus Segmentation Measurements to Quantify Wilms Tumor Volume on MRI in Pediatric Patients"

_cancers, 2023, doi:10.3390/cancers15072115_

Round 1
Reviewer 1 Report
Thanks for giving me an opportunity to review this paper. In this study, the authors investigated whether manual-segmentation-based volume measurements are more precise and explored whether deep learning can automate these segmentations. Several points need to address.
Abstract:
1. How many MRI images were used in this study?
2. Which deep learning model was used in this study?
Methods:
1. Provide more information about deep learning algorithms.
2. How many radiologists were included to review the MRI images
Results:
1. Show each fold performance
Discussion:
1. Provide more information about the clinical perspective.
Author Response
We thank the reviewer for his/her time and effort in reviewing our manuscript.We have addressed the comments point by point and have attached a document with the responses to all reviewers and the editor, since some of the comments overlap.
Reviewer 1:
Abstract:
- How many MRI images were used in this study?
We added the following statement in our abstract:
“We included the MRI images of 45 Wilms tumor patients (age 0-18 years).”
- Which deep learning model was used in this study?
We added the following statement in the abstract:
“Next, we created an automated segmentation method by training a nnU-Net in a 5-fold cross-validation.”
Methods:
- Provide more information about deep learning algorithms.
We added the following information:
“Tumor segmentation was automated using the deep learning application nnU-Net [7], which is a self-adapting framework for U-net based segmentation. Input to nnU-Net were full 3D T2-weighted images, output were binary segmentation maps.”
- How many radiologists were included to review the MRI images
One pediatric radiologist measured the radiological tumor volumes. Three independent observers segmented the tumors on MRI, under supervision of one pediatric radiologist. This was added to the methods
Results:
- Show each fold performance
We have added the results of the performance for each fold to the results section:
“When comparing the manual and automated segmentation for each segmented tumor (n=52), the median non-zero Dice was 0.90 (range: 0.03-0.97) (0.89 for fold 1, 0.9 for fold 2, 0.90 for fold 3, 0.95 for fold 4, 0.89 for fold 5, folds did not differ significantly). The median Hausdorf distance was 7.2 mm (5.5 for fold 1, 7.2 for fold 2, 11.3 for fold 3, 7.0 for fold 4, and 8.0 for fold 5. There was no significant difference between the different folds).”
Discussion:
- Provide more information about the clinical perspective.
We added the following statement in our discussion.
“We showed that clinically used volume measurements in Wilms tumor might not be accurate. Changing the method of volume measurements may well impact clinical decision making. Further research has to be conducted to investigate the clinical impact of our findings.”
Reviewer 2 Report
The submitted article is an interesting work and having novelty. I observed few updates are required in this work:
(1) The Introduction Section is well written but it is too short to understand by the readers. Also, the research contribution by the authors are not listed clearly. This is just a short discussion of literature review.
(2) Section 2.3.3 is about deep learning-based tumor volume measurements. This must have an algorithm or flowchart or architecture diagram for better understanding by the readers.
(3) Elaborate more about statistical approaches used in this work.
(4) Why Figure 4 is required here? Justify
(5) Conclusion section is too short for such an interesting and important topic.
(6) In Discussion section, mention the significance of this work.
(7) Comparative analysis is required in this work.
Author Response
We thank the reviewer for his/her time and effort in reviewing our manuscript.We have addressed the comments point by point and have attached a document with the responses to all reviewers and the editor, since some of the comments overlap.
Reviewer 2:
The submitted article is an interesting work and having novelty. I observed few updates are required in this work:
- The Introduction Section is well written but it is too short to understand by the readers. Also, the research contribution by the authors are not listed clearly. This is just a short discussion of literature review.
We added our contributions to the introduction:
“The aim of this study was twofold. Our first aim was to assess whether manual segmentation of MRI can accurately quantify the volume of Wilms tumors when compared to radiological volume measurements. Our second aim is to prove if manual segmentation can be automated using deep learning.”
- Section 2.3.3 is about deep learning-based tumor volume measurements. This must have an algorithm or flowchart or architecture diagram for better understanding by the readers.
A new figure has been added to the methods section, see also remark 1 from the editor.
- Elaborate more about statistical approaches used in this work.
We clarified our statistical analysis in section 2.4 “Statistical analysis”. See also remark 3 from the editor.
- Why Figure 4 is required here? Justify
We agree with the reviewer that Figure 4 does not add information to the results in the text and we therefore removed it as suggested by the reviewer.
- Conclusion section is too short for such an interesting and important topic.
We extended our conclusion to add more context and clarity:
” Our results show that currently used geometry-based radiological tumor volume measurements underestimate tumor volume compared to manual segmentation. This underestimation was about 10%, independent of tumor size. Deep learning shows potential to replace manual segmentation in this process to utilize the benefits of segmentation-based volume measurements while limiting time- and observer constraints. A change in volume measurement technique may have consequences for the stratification of treatment in individual children with a Wilms tumor.”
- In Discussion section, mention the significance of this work.
We have addressed this issue in the discussion, see also remark 2 of the editor.
- Comparative analysis is required in this work.
In the discussion a statement is added about the work of Müller et al, however this study population (17 WT patients) is too small for a comparative analysis, and there are no other studies currently available for a comparative analysis.
“Müller et al. found an average underestimation of 22% in Wilms tumor volumes when measured as an ellipsoid compared to human expert annotations (N = 17 patients) [10]. Our paper, which is the largest to date regarding the number of Wilms tumor patients, also shows an underestimation, and provides a deep learning framework to accurately quantify Wilms tumor volume.
Reviewer 3 Report
The authors compared radiological and segmentation measurements of size judgments in Wilms tumor, a typical pediatric tumor, and concluded that the radiological estimation may be an underestimate. The authors concluded that the radiological estimation may underestimate the size of Wilms' tumors.
Although there are no major problems with the purpose, method, or results, there is a lack of significance in why Wilms tumor is used in the development of these methods.
It would be fine to use Wilms as a starting point, but is this method intended to be used only for Wilms?
After all, other types of tumors should be considered as well. This may also help to identify issues in other parts of the body.
Author Response
We thank the reviewer for his/her time and effort in reviewing our manuscript.We have addressed the comments point by point and have attached a document with the responses to all reviewers and the editor, since some of the comments overlap.
Reviewer 3:
The authors compared radiological and segmentation measurements of size judgments in Wilms tumor, a typical pediatric tumor, and concluded that the radiological estimation may be an underestimate. The authors concluded that the radiological estimation may underestimate the size of Wilms' tumors.
- Although there are no major problems with the purpose, method, or results, there is a lack of significance in why Wilms tumor is used in the development of these methods.
Wilms tumor treatment is largely depended on volumetric assessment, but it is currently unknown of the clinically used measurements are accurate. Therefore, we started this work in this patient group. Besides this, we treat a relatively large number of patients with a Wilms tumor so we had access to the required number of patients.
We added the following sentence in our discussion:
“We conducted this research in WT patients because of the importance of volumetric assessment during treatment.”
- It would be fine to use Wilms as a starting point, but is this method intended to be used only for Wilms? After all, other types of tumors should be considered as well. This may also help to identify issues in other parts of the body
We thank the reviewer for this comment and agree with his point of view. We have added the following to our discussion:
“Although the automated segmentation method was developed specifically for Wilms tumor patients, the general methodology can easily be extrapolated to other tumor types. We hypothesize that our conclusion about the radiological volume measurements holds across tumor types. The developed automated segmentation method can easily be retrained for other tumor types, both in the pediatric as in the adult population. In future research, we will extend our framework to other pediatric tumor types such as neuroblastoma.
Reviewer 4 Report
Few suggestions to the authors to incorporate in the Manuscript to further improve the quality of presentation.
1. The introduction section should be further strengthened with atleast 15-18 recent studies included and the brief about the objective of those studies and observations made.
2. A short paragraph on the methods used in the study and need for the same explanation on the methods and novelty of the present study should also be presented at ending note of the Introduction section.
3. The discussion section should be further strengthened with a comparison of outcomes of previous studies to present study.
4. A Table showing summary of methods and outcomes previous studies and the present study to be included.
5. Table on histological criteria for wilms tumor classification can be added.
6. Table on criteria for decision making related to segmentation on how to stratify it as low, intermidiate, high risk
7. Is staging related discussion important to discuss and related protocols followed?
8. Mri Images can be marked for region of interest and explained.
9. What are the various Imaging techniques and it's diagnostics findings can be included.
10. Overall manuscript needs a comprehensive presentation of the discussion on the topic presented to further interest the outcomes.
Author Response
We thank the reviewer for his/her time and effort in reviewing our manuscript. We have addressed the comments point by point and have included a document with the answers to comments of alle reviewers and the editor since some of these overlap
Reviewer 4:
Few suggestions to the authors to incorporate in the Manuscript to further improve the quality of presentation.
- The introduction section should be further strengthened with atleast 15-18 recent studies included and the brief about the objective of those studies and observations made.
Most studies are focused on automated staging of Wilms tumor. These articles are out of scope for our article. We did emphasized in the discussion the work by Müller et al., which is focused on automated segmentation and applications of Wilms tumor on MRI.See also remark 7 of reviewer #2.
- A short paragraph on the methods used in the study and need for the same explanation on the methods and novelty of the present study should also be presented at ending note of the Introduction section.
we added the following statement to the introduction:
“Automated segmentation has the potential of utilizing the more precise manual segmentation-based volume measurements while limiting the time- and observer constraints.”
- The discussion section should be further strengthened with a comparison of outcomes of previous studies to present study.
See our responses to your second comment and to comment 7 of reviewer # 2.
- A Table showing summary of methods and outcomes previous studies and the present study to be included.
Only the study of Muller et al. focused on automated segmentation and volume measurements in Wilms tumor patients. We have added this reference to the manuscript. See also previous remarks.
- Table on histological criteria for wilms tumor classification can be added.
Histology is typically not known before surgery due to a risk of tumor rupture. This means that it does not influence treatment choices that are based on tumor volume. Therefore, histology is not included in this study. If required, we can add this explanation in the discussion.
- Table on criteria for decision making related to segmentation on how to stratify it as low, intermidiate, high risk
The stratification between low, intermediate, and high risk is mainly based on the pathology report of the tumor, whereas the volume measurements we describe are performed pre-operatively. The pre-operative size of the tumor is used in the UMBRELLA protocol for 2 decisions. i: Is NSS a possibility, and ii: if the size after pre-operative chemotherapy is >500 cc post-operative chemotherapy may be intensified. This is explained in the introduction. To further clarify this, we have added a table in the supplementary material that shows the addition of doxorubicin for larger tumors in intermediate-risk, stromal or epithelial type tumors, and refer to this table in the introduction.
- Is staging related discussion important to discuss and related protocols followed?
This manuscript focusses on volume measurements. Although volume measurements are part of Wilms tumor staging, as we discussed in our introduction and discussion, and can be seen in supplemental table 1, the whole staging protocol is out of scope for our article.
- Mri Images can be marked for region of interest and explained.
We added extra explanation in the caption of figure 4, explaining the manually labelled regions of interest more in depth.
“The reference manual segmentation (which outlines the tumor on MRI) is shown in purple, the deep learning segmentation is shown projected on the manual segmentation in blue. Note that in figure 3B these regions have an almost complete overlap.”
- What are the various Imaging techniques and it's diagnostics findings can be included.
We added a short statement about the place of MRI and CT in Wilms tumor research.
“In the International Society of Paediatric Oncology-Renal Study Tumor Group (SIOP-RTSG) UMBRELLA guidelines MRI is preferred because of the absence of a radiation dose and the excellent soft tissue contrast.”
- Overall manuscript needs a comprehensive presentation of the discussion on the topic presented to further interest the outcomes.
We added the following paragraphs to our article. First in our introduction:
“The aim of this study was twofold. Our first aim was to assess whether manual seg-mentation of MRI can accurately quantify the volume of Wilms tumors. Our second aim is to prove if manual segmentation can be automated using deep learning. Automated segmentation has the potential of utilizing the more precise manual segmentation-based volume measurements while limiting the time-and observer constraints.”
And in our discussion:
“Besides volume measurements, other factors are of importance in staging and risk stratification of Wilms tumor. Ma et al. showed that they could accurately predict WT stage 1 on CT. Research on automatic staging of WT on MRI is needed [11].”
“We conducted this research in WT patients because of the importance of volumetric assessment during treatment. Although the automated segmentation method was developed specifically for Wilms tumor patients, the general methodology can easily be extended to other tumor types. We hypothesize that our conclusion about the radiological volume measurements hold acro ss tumor types. The developed automated segmentation method can easily be retrained for other tumor types, both in the pediatric as in the adult population. In future research, we will extend our framework to other pediatric tumor types such as neuroblastoma “
“We showed that clinically used volume measurements in Wilms tumor are not ac-curate. Changing the method of volume measurements might impact clinical decision making. Further research has to be conducted to investigate the clinical impact of our findings.”
Round 2
Reviewer 2 Report
Authors have addressed my comments positively
Author Response
Reviewer 2:
Authors have addressed my comments positively.
Thank you.
Reviewer 3 Report
The authors responded to the reviewers' comments and submitted a revision.
But it's not clear why we're starting with Wilms Tumor on this subject. Even if we started with WT, if we were to add data from other solid tumors, it would still be sufficiently persuasive, but at present the value of the results would be insufficient.
Author Response
We thank the editor and the reviewers for the second round of feedback. We have addressed all points and changed the manuscript accordingly.
Reviewer 3:
The authors responded to the reviewers' comments and submitted a revision.
But it's not clear why we're starting with Wilms Tumor on this subject. Even if we started with WT, if we were to add data from other solid tumors, it would still be sufficiently persuasive, but at present the value of the results would be insufficient
In Wilms tumor treatment, volumetric assessment if of higher importance than in other pediatric tumors. Therefore, we wanted to investigate the limitations of currently used volume measurements. Furthermore it is the largest group of solid tumor patients in our hospital so this new method will have clinical implications rather quickly. We will extend our experience to other patient groups. We have clarified this further in the discussion paragraph:
The differences between manual segmentation and radiological volume measurements may apply for any different types of tumor. We decided to conduct this research on WT patients first because of the importance of volumetric assessment during treatment. Furthermore, in our hospital the solid tumor diagnosed the most is WT. So this new method of volume measurements will have clinical implications rather quickly.
Reviewer 4 Report
The authors have well addressed the comments. I recommend the authors to further provide an comprehensive conclusion with more details on the outcomes . Presently the conclusion is too short.
Also, the authors should avoid using we did, we considered. Instead it can be rephrased to "in the present study" - Kindly change it throughout.
English language and style check is required.
Author Response
We thank the editor and the reviewers for the second round of feedback. We have addressed all points and changed the manuscript accordingly.
Reviewer 4:
The authors have well addressed the comments. I recommend the authors to further provide an comprehensive conclusion with more details on the outcomes . Presently the conclusion is too short.
We have extended the conclusion:
Our results show that currently used geometry-based radiological tumor volume measurements by review radiologists underestimate tumor volume in Wilms Tumor compared to manual segmentation. This underestimation was about 10%, independent of tumor size. A change in volume measurement technique may have consequences for the stratification of treatment in individual children with a Wilms tumor, and thus have clinical implications.
Deep learning shows potential to replace manual segmentation in this process to utilize the benefits of segmentation-based volume measurements while limiting time- and observer constraints. This method of volume measurements using manual or semi-automated segmentation may be extrapolated to other (pediatric) tumor types.
Also, the authors should avoid using we did, we considered. Instead it can be rephrased to "in the present study" - Kindly change it throughout.
Since there is no recommended styleguide listed in the guidelines to authors, we chose the AMA guidelines throughout our paper: The American Medical Association Guide for Authors and Editors promotes the active voice: “In general, authors should use the active voice” https://www.amamanualofstyle.com/view/10.1093/jama/9780190246556.001.0001/med-9780190246556-chapter-7-div2-296
Changing this however is possible if needed.
English language and style check is required.
The manuscript has been checked by a native speaker and was corrected accordingly.

Round 3
Reviewer 3 Report
Now, I have no point to be altered.